

# Age trends of genetic parameters and genotype-by-environment interactions for growth traits of *Eucalyptus urophylla* clones in South-China

Guangyou Li[1], Zhaohua Lu[1], Deming Yang[1], Yang Hu[2] and Jianmin Xu[1]

[1] Key Laboratory of State Forestry and Grassland Administration on Tropical Forestry, Research Institute of Tropical Forestry, Chinese Academy of Forestry, Guangzhou, China
[2] Xinhui Research Institute of Forestry Science, Jiangmen, China

Corresponding authors
Guangyou Li, luosuo3000@163.com
Jianmin Xu, jianmxu@163.com

## ABSTRACT

*Eucalyptus urophylla* S.T. Blake, an important economic tree species, is widely cultivated as a raw material source for pulpwood, veneer plywood, and sawlog timber in southern China. As a tree in multiple environments, tree-breeding programs can assess genotype by environment (G × E) interactions and identify the suitable genotype for a specific environment. G × E interactions related to growth traits and soil factors have not been adequately studied for clones of *Eucalyptus urophylla* and its hybrids. To examine this important question, trials containing 20 clones of *E. urophylla* and its hybrids were established at three sites in southern China: Shankou (SK), Tiantang (TT), and Xiniujiao (XNJ). These sites each have different soil conditions but similar geographical and climatic conditions. With the data across nearly eight years, average phenotypic trends and broad sense repeatability (H2) were modeled, G×E interactions between clones and diverse soil environments were estimated, genetic gains of clones were calculated, and the adaptabilities of *E. urophylla* clones in different soil environments were compared. Average survival trends for clones tended to show a moderate decrease while growth traits tended to show sharp increases with age. At the same age, sites were ordered for average survival and growth traits as TT>SK>XNJ while H2 values for growth traits by site followed the basic order TT>SK>XNJ. The H2 values for growth traits at SK tended to increase at first, platform, and then smooth with age. The H2 values for growth traits at TT were high and stable across ages, and those at XNJ tended to undulate largely at a relatively low level across ages. Genetic correlations for growth traits between any pair of sites tended to increase at first and then decrease. A genetic correlation was strong between SK and TT, intermediate between SK and XNJ, and weak between TT and XNJ. It was concluded that: (1) clones tended to be adapted better to an environment with acidic and loamy soil with a clay content of about 45.6%, the soil depth from the surface to parent material about 1.5 m, and the previous vegetation of *Eucalypts*. (2) The G×E interactions between clones and sites are weaker if the environmental conditions between the sites are similar, and which are stronger if the environmental conditions between the sites are different. (3) The optimum selection age for clones ranged from 1.5 to 3.5 years old, while the optimum selection growth trait is individual tree volume.

# INTRODUCTION

*Eucalyptus*, a widely cultivated hardwood genus in tropical and subtropical regions worldwide, most of its species originate from Australia. Since 1980, *Eucalyptus urophylla* S.T. Blake (*E. urophylla*) has been widely cultivated as a source of the raw material for pulpwood, veneer plywood, and sawlog timber in southern China because of its adaptability, rapid growth rate, and versatile wood properties (*Liang et al., 2022*). Unfortunately, current *Eucalyptus* plantations in China are based largely on the available land resources rather than on biological patterns of genotype-by-environment (G× E) interactions. As a result, it is difficult to deliver optimal genetic gains across the plantation estates.

In the past 50 years, breeding and selection programs have been increasingly implemented on various *Eucalypts* including *E. urophylla* in the corresponding plantation country. The goals of many breeding programs for *E. urophylla* have been to breed and select more well-adapted, high-yielding clones (*Osorio, White & Huber, 2001*; *Xu et al., 2004*), to enhance plantation production for commercial purposes (*Prado et al., 2019*), and develop clones resistant to abiotic and biotic stress (*Gan et al., 2004*; *Mangwanda, Myburg & Naidoo, 2015*; *Rezende et al., 2019*).

An important silvicultural principle is to plant trees that are well adapted to the environmental conditions of target sites, which is also a crucial guideline for tree breeding and selection. The basic theory of planting trees according to the plantation environments is based on G × E interactions, which arise when genetic entities differ in their responses to an array of environmental conditions. The main effects of G × E interactions can significantly affect growth traits, such as plant height and diameter at breast height, and secondary metabolite production (*Zhou et al., 2021*; *Campbell et al., 2023*). The response patterns of G × E interactions involve changes in the ranking of genotypes and alterations in scale. When the former type of G × E interaction presents at the provenance, family, or clonal level, it is of particular importance to tree breeders because it affects critical decisions for developing an optimal breeding strategy and for enhancing realistic genetic gains that can be realized in plantations (*Wang et al., 2016*; *Liu et al., 2024*; *Liu et al., 2023*). Genotypes that adapt well to a specific site or perform well across many sites can be developed and propagated by breeders or growers (*Chipeta et al., 2017*).

Although some studies reported the age trends of phenotypic and genetic parameters of *E. urophylla*, the growth traits from two tree ages may not be enough to reveal the age patterns (*Pinto et al., 2014*; *Berg et al., 2016*). To our knowledge, few references reported the trends of phenotypic and genetic parameters, and genetic and environmental correlations for growth traits in *E. urophylla* and clones of hybrids involving this species using data from multiple sites and many ages. Only a few reports on the G × E interactions of *E. urophylla* or its hybrids (*Wu et al., 2012*; *Silva et al., 2018*; *Berg et al., 2016*). For other Eucalypts species, there are some reports on *E. globulus*, *E. grandis* and *E. pellita* (*Endo & Wright, 1993*). However, all of these studies focused on G × E interactions between families, or clones, and geographical (*e.g.*, elevation) and meteorological (*e.g.*, rainfall) factors (*Lambeth, Endo & Wright, 1994*; *Mckeand et al., 2006*; *Apiolaza, 2012*). G × E interactions between families, or clones, and soil factors in *E. urophylla* and its hybrid clones were unknown. *Li et al.*

**Table 1  Geographic and meteorological information on the three experimental trial sites.**

| Location | Latitude | Longitude | Elevation (m) | MAT (°C) | MAR (mm) | MAEC (mm) | Replicates | Clones |
|----------|----------|-----------|---------------|----------|----------|-----------|------------|--------|
| **SK** | 21°35′N | 109°42′E | 35 | 23.1 | 1,594 | 1,133 | 10 | 20 |
| **TT** | 21°52′N | 109°21′E | 47 | 20.0 | 1,802 | 1,225 | 10 | 20 |
| **XNJ** | 21°44′N | 108°44′E | 25–35 | 22.5 | 2,104 | 1,340 | 10 | 20 |

**Notes.**

Note: MAT is the mean annual temperature, MAR is the mean annual rainfall, and MAEC is the mean annual evaporation capacity. The meteorological data were supplied by the National Data Center for Meteorological Sciences (http://data.cma.cn/), China.

*(2015)* found that G × E interactions for growth and foliar nutrient concentrations in radiata pine were significantly associated with the soil nutrient levels of nitrogen and total phosphorus levels. Additionally, a range of other studies had found that some growth traits and wood properties for *E. urophylla* and *E. grandis* were affected by the soil textures (*Xu et al., 2002*; *Henri, 2001*; *Gava & Goncalves, 2008*; *Stape et al., 2010*).

The main objectives of this current study were to use the genetic parameters for growth traits at three sites at ages ranging up to across 7.9 years to (1) model the age trends of phenotype and broad sense repeatability (H2); (2) estimate genetic correlations and discuss G × E interactions between clones and environments (mainly soil factors, climate factors, and biotic factors) at different ages; and, (3) compare the adaptability of *E. urophylla* and its hybrids clone in various environmental conditions and calculate the genetic gains to select the top five clones across nearly eight years. Our results may provide useful insights into silviculture and tree breeding strategies in the optimum selection age, environmental conditions, and a preferred selection index for growth traits.

# MATERIALS & METHODS

## Experimental sites

Three trials were established in Guangxi Zhuang Autonomous Region, China: Shankou (SK) Forestry Farm (21°35′N, 109°42′E), Shankou Township, Hepu County of Beihai City; Tiantang (TT) Village, Qinlian Forestry Farm (21°52′N, 109°21′E), Hepu County of Beihai City; and, Xiniujiao (XNJ) (21°44′N, 108°44′E) Township in Qinzhou City. All three trial sites were within 9 to 10 kilometers of the coast sea (straight-line distance). Details on the geography and climate of these three sites were acquired from local county weather bureaus and presented in Table 1. The geographical and climatic conditions for the three trial sites are very similar. The three sites have altitudes between 25–50 m, mean annual temperatures of 20.0–22.3 °C, mean annual rainfalls of 1,594–2,104 mm and mean annual evaporation of around 1,133–1,340 mm, and all three sites are affected by south subtropical monsoons in spring-summer.

The soils at TT and XNJ sites are calcareous soils developing from basalt according to the parental materials, and the soil at the SK site is epeiric sea sedimentary soil, furthermore, the soils at SK, TT, and XNJ are aeolian sandy, lime, and phosphor calcic soils, respectively. Based on the soil texture, however, the soil at SK is sandy, with a clay content of about 260 g/kg, the soil at TT is loamy soil with a clay content of 456 g/kg, and the soil at XNJ is

**Table 2  Soil nutrient information at the three experimental trial sites.**

| Location | pH value | Organic matter (g/kg) | Total N (g/kg) | Total P (g/kg) | Total K (g/kg) | Hydrolytic N (mg/kg) | Available P (mg/kg) | Available K (mg/kg) | Clay content (g/kg) | Soil thickness (g/kg) |
|---|---|---|---|---|---|---|---|---|---|---|
| SK | 6.7 | 13.2 | 0.7 | 0.2 | 3.4 | 76.4 | 0.8 | 16.8 | 260 | 2 |
| TT | 5.2 | 42.1 | 1.5 | 0.4 | 4.3 | 111 | 1.1 | 22.3 | 456 | 1.5 |
| XNJ | 7.8 | 15.3 | 0.9 | 0.2 | 13.7 | 106.7 | 21.2 | 53.5 | 705 | 0.5 |

clay soil in which clay content is about 705 g/kg. Some years ago, severe soil erosion had occurred at XNJ due to a long period of inappropriate plow and cultivation and this has resulted in a thin soil in which the average depth is about 0.30–0.50 m from surface to bottom of horizon. One soil sample of about 500 g was collected at the actual depth from 0 to 0.50 m according to the five-point sampling mode at each site in April 2011.

Information on key soil nutrients and properties at each of the sites are presented in Table 2, based on one mixed homogeneously sample and measurements. Obviously, soils at the three trial sites were lower in nitrogen, phosphorus, and potassium. Even so, the pH values at the three sites differed markedly with values of 6.7 at SK, 5.2 at TT, and 7.8 at XNJ, indicating weakly acidic soil at SK, acidic soil at TT, and alkaline soil at XNJ. The average depth from the surface to parent material at three sites is about 2 m at SK, 1.5 m at TT, and 0.5 m at the XNJ site. The dominant understorey vegetation within the trials consisted mainly of *Leucas mollissima* Wall. Var. Chinensis Benth., *Rhodomyrtus tomentosa*, *Dicranopteris dichotoma* Bernh., and other thorn shrubs. Previous vegetation was a plantation of *E. urophylla* and *E. tereticornis* natural hybridization progeny at SK and TT site, and a plantation of *Pinus massoniana* Lamb. at the XNJ site.

At each trial site, planting pits (50 cm long, 50 cm wide, and 40 cm high) had been dug in rows along contour lines. In each pit 0.25 kg superphosphate and 0.25 kg of a special eucalypt compound fertilizer mix (N:P: K =15:15:5) were applied before trees were planted to ensure every individual tree had adequate nutrients for at least the first three months after planting.

## Genotypes

Twenty clones were planted at the three trial sites in April 2003. These had been bred or selected at the Research Institute of Tropical Forestry of the Chinese Academy of Forestry, the Leizhou Forestry Bureau of Guangdong Province in China, and the Dongmen Forestry Farm of Guangxi Province in China. These clones could represent five taxa: *E. urophylla*, *E. urophylla* × *E. grandis*, *E. urophylla* × *E. camaldulensis*, *E. urophylla* × *E. tereticornis* and *E. leizhouensis* No. 1× *E. urophylla*. Details of the 20 clones are presented in Table 3.

## Methods

The field experiments were designed in a randomized complete block with 10 replications with each clone represented by a row plot of five trees in each replicate. Plots were aligned along the contour line with a between-tree spacing of 2 m, and the distance between rows was 4 m. Thus, the trials were established with initial stockings of 1,250 trees per hectare.

**Table 3  Information regarding *E. urophylla* clonal sources used at the three experimental trial sites.**

| No. | Genotypes | No. | Genotypes | No. | Genotypes | No. | Genotypes |
|-----|-----------|-----|-----------|-----|-----------|-----|-----------|
| 1 | U | 6 | U | 11 | U × G | 16 | U × G |
| 2 | U | 7 | U × T | 12 | U × G | 17 | U × C |
| 3 | U | 8 | U | 13 | U × G | 18 | U × G |
| 4 | U | 9 | U | 14 | U × G | 19 | U* |
| 5 | U | 10 | U × T | 15 | U × G | 20 | L₁ × U |

**Notes.**

Note: U represents *E. urophylla*, G represents *E. grandis*, C represents *E. camaldulensis*, L1 represents *E. leizhouensis* No.1, and T represents *E. tereticornis*. * indicates the control clone U6 (a commercial clone of *E. urophylla*).

## Data collection

Growth traits, including total height (H in m), diameter at breast height over bark (DBH in cm), and survival were assessed at 6 (0.5 years), 12 (1 year), 18 (1.5 years), 30 (2.5 years), 42 (3.5 years), 56 (4.7 years), 68 (5.7 years) and 95 (7.9 years) months after the seedlings were planted. H was assessed using a hypsometer and DBH was measured with a diameter tape. Individual tree volume over bark (V in m³) was calculated using the following equation (*Wu et al., 2011*; *He et al., 2012*; *Hodge & Dvorak, 2015*):

$$V = H \times DBH^2 / 30,000 \tag{1}$$

## Statistical analysis

The H, DBH and V values per clone for each year (ranged from 0.5 to 7.9 years) on each site were collected and applied to a linear model (*Hansen & Roulund, 1996*; *Zhu, 2000*; *Wang, 2006*):

$$Y_{ijkl} = \mu + S_i + B_{j(i)} + F_k + SF_{ik} + BF_{jk} + SBF_{iik} + e_{ijkl}, \tag{2}$$

where *i, j, k* and *l* indicate the observed numbers of sites, blocks, clones, and plots, respectively,

$Y_{ijkl}$ is the performance of the ramet of the kth clone of the lth plot in the *j*th block within the *i*th site,

$\mu$ is the overall mean of the experimental trial, $S_i$ is the fixed effect of the *i*th site,

$B_{j(i)}$ is the fixed effect of the jth block within the *i*th site,

$F_k$ is the random effect of the *k*th clone,

$SF_{ik}$ is the random effect of the interaction between the *k*th clone and the *i*th site,

$BF_{jk}$ is the random effect of the interaction between the *j*th block and the *k*th clone,

$SBF_{jik}$ is the random effect of the interaction between the *k*th clone and the *j*th block within the *i*th site, and

$e_{ijkl}$ is the random error of the lth individual in the kth clone within the *j*th block of the *i*th site.

Clone repeatability was calculated with the equation of broad sense heritability of family, using the following equation (*Xu, 1988*):

$$H_c^2 = \sigma_c^2 / (\sigma_c^2 + \sigma_e^2 / r), \tag{3}$$

The single tree's repeatability was estimated with the formula of broad sense heritability, which was calculated using the following formula (*Xu, 1988*):

$$H_s^2 = 4\sigma_c^2 / (\sigma_c^2 + \sigma_e^2), \tag{4}$$

where $\sigma_c^2$ and $\sigma_s^2$ indicate the variance of the clone and the single tree; $\sigma_e^2$ indicates the error of the variance; and $r$ indicates the number of replicates.

Genetic gain was calculated using the following equation (*Huang & Xie, 2001*):

$$\Delta G = H_c^2 \times S \times X^{-1} \tag{5}$$

$$S = (x - X), \tag{6}$$

where $S$ is the skewness between the mean of the superior clones and that of the control clones, $X$ is the mean value of the control clone, x is the mean value of the superior clones, and $h_c^2$ is equivalent to clone repeatability (broad sense heritability or $H_c^2$) of the clones.

Variances of growth traits for H, DBH, and V across seven assessment ages at the three experimental sites were analyzed using GLM procedures in the SAS statistical software (*SAS Institute Inc, 1990*; *Huang & Xie, 2001*). Due to the unbalanced data, type-B genetic correlation parameters and genetic gains were estimated using GLM procedures in the SAS statistical software (*Huber, White & Hodge, 1994*; *Huang & Xie, 2001*). Curves of means of growth trait across ages at the three sites were simulated using SPSS statistics 21.0 (SPSS ver. 21.0; Armonk, NY, USA), with the equation used to describe the growth trends being selected based on maximizing R2, lower sums of mean squares residual errors, and standard errors.

## RESULTS

### Average trends of survival and growth traits

The trends in survival rates (%) from 0.5 to 7.9 years old at the three sites were similar (Table 4), with a moderate decrease from 96.9 to 72.5, from 94.8 to 77.6, and from 75.3 to 52.4 at SK, TT, and XNJ, respectively. Significant differences in survival rates were observed among the tree ages at each site. The survival rates reached a plateau from the beginning of 4.7 years old at XNJ. However, survival rates at SK and TT tended to be stable levels from 5.7 to 7.9 years old. Interestingly, the average survivals at SK and TT were similar and significantly higher than that of XNJ at every tree age.

The H at the three sites increased with age across nearly eight years, which can be expressed for SK, TT, and XNJ using the following equations, respectively:

$y_1 = 7.0973\ln(x) + 5.4412$, $R^2 = 0.9685$;
$y_2 = 7.3883\ln(x) + 5.8401$, $R^2 = 0.9717$; and
$y_3 = 2.2964\ln(x) + 2.5609$, $R^2 = 0.9276$.

The average H growth of *E. urophylla* clones at SK, TT, and XNJ increased sharply from 0.5 to 3.5 years old and then increased smoothly in the following years (Table 5). Significant differences in H growth were found among the tree ages at each site. Moreover, the monthly mean increments of H growth reached the highest levels at the three sites

**Table 4** Average trends for survival (%) of *E. urophylla* clones across eight years at the three trial sites.

| Age | Mean | | |
|---|---|---|---|
| | SK | TT | XNJ |
| 0.5 | 96.9±9.43Aa | 94.8±12.25Aa | 75.3±22.78Ba |
| 1.0 | 96.2±9.90Aa | 94.4±12.56Aab | 72.5±22.97Ba |
| 1.5 | 94.7±12.44Aa | 92.1±16.70Aabc | 69.2±23.10Bab |
| 2.5 | 93.4±13.76Aab | 90.3±19.96Aabc | 63.3±23.48Bbc |
| 3.5 | 90.1±18.19Ab | 88.0±22.05Acd | 62.3±23.09Bc |
| 4.7 | 85.5±19.77Ac | 86.8±21.70Ad | 58.0±23.64Bcd |
| 5.7 | 75.1±25.58Ad | 79.2±24.91Ae | 56.5±22.75Bcd |
| 7.9 | 72.5±26.26Ad | 77.6±24.73Ae | 52.4±22.03Bd |

Notes.

Note: Mean is the average survival (%). SK represents the Shankou site, TT represents the Tiantang site and XNJ represents the Xiniujiao site. Different capital letters indicated significant differences in the average survival among the sites in the same row at the 0.05 levels. Different small letters indicated significant differences in the average survival among the tree ages in the same column at the 0.05 levels.

**Table 5** Average trends for height of *E. urophylla* clones across eight years at the three trial sites.

| Age | Mean H (m) | | |
|---|---|---|---|
| | SK | TT | XNJ |
| **0.5** | 2.28 ± 0.18Ag | 2.27±0.28Af | 2.17±0.21Bh |
| **1.0** | 4.10 ± 0.44Af | 3.58±0.48Bf | 3.40±0.37Cg |
| **1.5** | 7.92 ± 1.05Ae | 7.44±0.96Be | 6.54±0.76Cf |
| **2.5** | 11.1 ± 1.61Bd | 11.92±1.74Ad | 8.93±1.12Ce |
| **3.5** | 15.26 ± 2.33Bc | 16.38±2.50Ac | 11.92±1.55Cd |
| **4.7** | 17.76 ± 1.93Ab | 18.03±3.47Ab | 15.46±1.24Bc |
| **5.7** | 18.60 ± 1.51Ab | 18.11±1.99Bb | 16.62±1.15Cb |
| **7.9** | 20.02 ± 1.36Aa | 20.35±2.08Aa | 18.15±1.03Ba |

Notes.

Note: SK represents the Shankou site, TT represents the Tiantang site and XNJ represents the Xiniujiao site. Different capital letters indicated significant differences in the average height among the sites in the same row at the 0.05 levels. Different small letters indicated significant differences in the average height among the tree ages in the same column at the 0.05 levels.

during 1−1.5 years old. The H growth of SK and TT was remarkably higher than that of XNJ during the investigation. The H growth of *E. urophylla* clones at the three sites at 4.7 years (around one rotation) varied from 15.46 to 18.04 m. DBH over bark across nearly eight years at the three sites increased with age, which can be expressed for SK, TT, and XNJ with the following equations, respectively:

$y_4 = 4.9138\ln(x) + 4.5095, R^2 = 0.9973;$

$y_5 = 5.4723\ln(x) + 4.2311, R^2 = 0.9917;$ and

$y_6 = 4.7483\ln(x) + 4.4367, R^2 = 0.9914.$

The average DBH growth of *E. urophylla* clones increased rapidly with age up to 2.5 years old and slowly from 2.5 to 7.9 years old (Table 6). An obvious difference in DBH growth at each site over these years. The DBH growth showed a remarkable difference among the three sites from 0.5 to 5.7 years old. Additionally, the average monthly increments in DBH growth exhibited the maximum during 1–1.5 years old at TT and XNJ, while its peak value

**Table 6 Average trends for DBH over the bark of *E. urophylla* clones across eight years at the three trial sites.**

| Age | Mean D (cm) | | |
|---|---|---|---|
| | SK | TT | XNJ |
| 0.5 | 1.25±0.24Bh | 1.20±0.33Bg | 1.34±0.22Ag |
| 1.0 | 4.04±0.28Ag | 3.35±0.41Cf | 3.65±0.37Bf |
| 1.5 | 6.80±0.36Af | 6.14±0.62Ce | 6.61±0.75Be |
| 2.5 | 9.04±0.81Be | 9.29±1.30Ad | 9.36±1.27Ad |
| 3.5 | 10.47±1.03Cd | 10.73±1.70Bc | 11.30±0.96Ac |
| 4.7 | 11.92±0.80Cc | 12.23±2.0B7b | 12.77±1.13Ab |
| 5.7 | 12.36±1.01Bb | 12.84±1.77Ab | 13.15±1.02Ab |
| 7.9 | 14.75±1.12Aa | 14.70±2.09Aa | 14.55±1.06Aa |

Notes.
Note: SK represents the Shankou site, TT represents the Tiantang site and XNJ represents the Xiniujiao site. Different capital letters indicated significant differences in the average DBH over the bark among the sites in the same row at the 0.05 levels. Different small letters indicated significant differences in the average DBH over the bark among the tree ages in the same column at the 0.05 levels.

**Table 7 Average trends for individual tree volume over the bark of *E. urophylla* clones across eight years at the three trial sites.**

| Age | Mean v (m³) | | |
|---|---|---|---|
| | SK | TT | XNJ |
| 0.5 | 0.00016±7.025E−05Bf | 0.00015±8.77E−05Bd | 0.00022±8.5E−05Ag |
| 1.0 | 0.00245±0.00054Af | 0.00147±0.00050Cd | 0.00181±0.00056Bg |
| 1.5 | 0.012670.00271Aef | 0.00992±0.00299Cd | 0.01130±0.00308Bf |
| 2.5 | 0.03239±0.00896Be | 0.03667±0.01434Ac | 0.03198±0.00956Be |
| 3.5 | 0.06048±0.01833Bd | 0.06661±0.02767Ac | 0.05738±0.01518Bd |
| 4.7 | 0.09083±0.01812Bc | 0.09493±0.04497Ab | 0.09203±0.02075Bc |
| 5.7 | 0.12186±0.08243Ab | 0.10420±0.03804Ab | 0.10562±0.02032Ab |
| 7.9 | 0.16092±0.02903Aa | 0.15550±0.05842Aa | 0.13685±0.02371Ba |

Notes.
Note: SK represents the Shankou site, TT represents the Tiantang site and XNJ represents the Xiniujiao site. Different capital letters indicated significant differences in the average individual tree volume over the bark among the sites in the same row at the 0.05 levels. Different small letters indicated significant differences in the average individual tree volume over the bark among the tree ages in the same column at the 0.05 levels.

of SK was from 0.5 to 1.0 years old. The DBH growth of *E. urophylla* clones at 4.7 years (about one rotation) differed significantly by the sites, with the highest in XNJ (12.77 cm), followed by TT (12.23 cm) and SK (11.92 cm).

For V over bark, the average trends over nearly eight years at the three sites followed a consistent pattern (Table 7). The curves for average V at SK, TT, and XNJ can be expressed with the following equations, respectively:

$y_7 = 0.0215x - 0.0158$, $R^2 = 0.9921$;

$y_8 = 0.0235x - 0.0178$, $R^2 = 0.9906$; and

$y_9 = 0.0201x - 0.0141$, $R^2 = 0.9849$.

The average V growth increased moderately with age from the beginning. There were striking differences in V growth at each site over these years. Apart from 5.7 years old, the
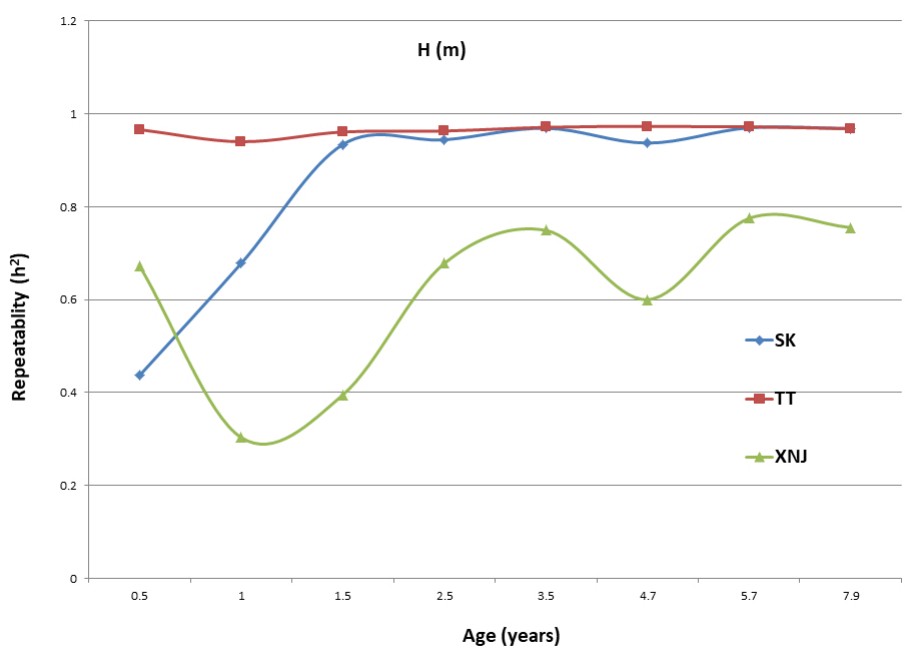

**Figure 1** $H_c^2$ Trends of *E. urophylla* clones for height at three sites across eight years.

V growth of the three sites showed a significant difference at the investigation ages. For individual tree volume over bark, the growth of *E. urophylla* clones at the three sites at 4.7 years (about one rotation) varied from 0.0920 (XNJ) to 0.1002 m³ (TT).

## Trends in $H_c^2$ for growth traits

Trends in broad sense heritability ($H_c^2$) for all the growth traits (H, DBH over bark, and V) followed consistent patterns (Fig. 1, Fig. 2, and Fig. 3). The values of all $H_c^2$'s at TT were dramatically greater than those at SK or XNJ, and those at SK were greater than those at XNJ. $H_c^2$ curves for growth traits across ages at TT were relatively stable, except for a slight increase at early ages. The $H_c^2$'s at SK increased at early ages significantly and after a short plateau, started to fluctuate slightly with age to 7.9 years old. In contrast, $H_c^2$'s at XNJ undulated largely at a relatively low level from 0.5 to 7.9 years old.

For H, the $H_c^2$ for *E. urophylla* clones varied from 0.437 to 0.970 at SK, 0.940 to 0.973 at TT, 0.303 to 0.775 at XNJ. For DBH over bark, the $H_c^2$ varied from 0.452 to 0.962 at SK, 0.854 to 0.972 at TT, and 0.250 to 0.801 at XNJ. For V, the $H_c^2$ varied from 0.564 to 0.942, 0.915 to 0.985, and 0.200 to 0.732 at SK, XNJ and TT, respectively.

## Genetic correlations

Genetic correlations among the same growth traits (H, DBH over bark and V) at the same age were assessed, pairwise, between the sites. For the growth traits, the peak values of genetic correlations were between SK and TT at each age (Table 8). The genetic correlations for the growth traits between pairs of sites increased at early ages, then decreased, and finally increased again with age, except for V between SK and TT, which increased at early ages before starting to decrease with age. The strongest genetic correlations between two sites

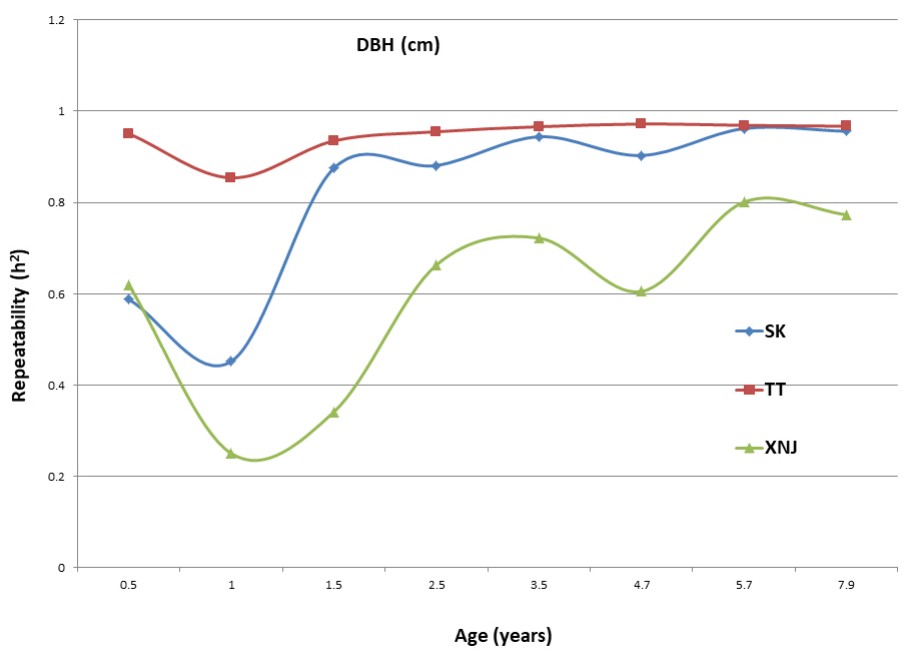

**Figure 2** $H_c^2$ Trends of *E. urophylla* clones for DBH over bark at three sites across eight years.

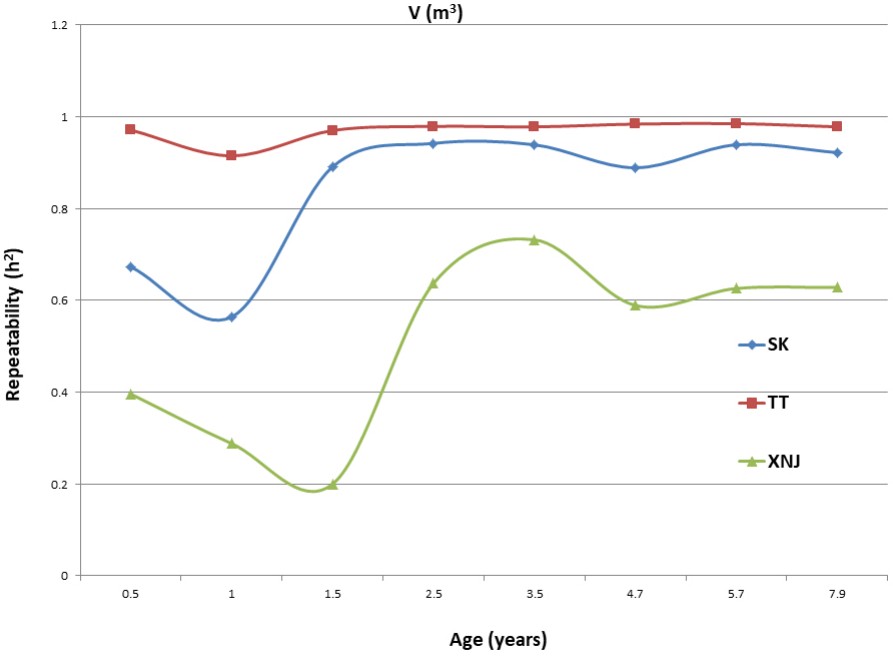

**Figure 3** $H_c^2$ Trends of *E. urophylla* clones for individual tree volume over bark at three sites across eight years.

**Table 8  Genetic correlation matrices of clone effects for growth traits at eight years between pairs of trial sites.**

| Traits | Age (yr) | 0.5 | | 1.5 | | 2.5 | | 3.5 | | 4.7 | | 5.7 | | 7.9 | |
|---|---|---|---|---|---|---|---|---|---|---|---|---|---|---|---|
| | Site | SK | TT | SK | TT | SK | TT | SK | TT | SK | TT | SK | TT | SK | TT |
| **Height** | TT | 0.75** | | 0.92** | | 0.96** | | 0.92** | | 0.88** | | 0.90** | | 0.93** | |
| | | (0.13) | | (0.00) | | (0.00) | | (0.00) | | (0.01) | | (0.00) | | (0.00) | |
| | XNJ | 0.21** | 0.52** | 0.35** | 0.52** | 0.43** | 0.57** | 0.38** | 0.52** | −0.09* | 0.26** | −0.18** | −0.15** | 0.07 | 0.01 |
| | | (0.37) | (0.29) | (0.30) | (0.27) | (0.27) | (0.25) | (0.29) | (0.27) | (0.38) | (0.36) | (0.33) | (0.34) | (0.38) | (0.38) |
| **DBH** | TT | 0.71** | | 0.85** | | 0.90** | | 0.90** | | 0.89** | | 0.89** | | 0.91** | |
| | | (0.14) | | (0.07) | | (0.00) | | (0.00) | | (0.01) | | (0.00) | | (0.00) | |
| | XNJ | 0.25** | 0.26** | 0.43** | 0.64** | 0.35** | 0.56** | 0.22** | 0.49** | −0.10* | 0.32** | −0.20** | −0.18** | 0.06 | −0.02 |
| | | (0.31) | (0.21) | (0.31) | (0.25) | (0.32) | (0.28) | (0.36) | (0.32) | (0.41) | (0.38) | (0.33) | (0.33) | (0.37) | (0.37) |
| **Volume** | TT | 0.76** | | 0.95** | | 0.94** | | 0.90** | | 0.88** | | 0.87** | | 0.85** | |
| | | (0.11) | | (0.00) | | (0.00) | | (0.00) | | (0.02) | | (0.07) | | (0.04) | |
| | XNJ | 0.34** | 0.54** | 0.78** | 0.74** | 0.65** | 0.72** | 0.60** | 0.71** | 0.22** | 0.53** | 0.09* | 0.14** | 0.30** | 0.23** |
| | | (0.26) | (0.21) | (0.18) | (0.19) | (0.22) | (0.21) | (0.24) | (0.22) | (0.39) | (0.34) | (0.35) | (0.34) | (0.33) | (0.34) |

**Notes.**

** and * indicates significant differences at the 0.01 and 0.05 levels, respectively. Standard errors are in brackets. SK represents the Shankou site, TT represents the Tiantang site and XNJ represents the Xiniujiao site.

in H, DBH over bark, and V were found in 2.5 years (0.96), 7.9 years (0.91), and 1.5 years (0.95), respectively.

## Genetic gains

Estimated mean genetic gains for H, DBH over bark, and V, as would be achieved by selecting the top five of the 20 clones at the three sites, increased with age at first and then decreased after peaking between 1.5 and 4.7 years of age (Table 9). The highest values of genetic gains at the same age were reached at the TT site.

## DISCUSSION

The average survivals at 5.7 years old at SK, TT, and XNJ were 75.1%, 79.2%, and 56.5%, respectively, which were similar to that of previous studies of *E. urophylla* (*Kien et al., 2009*; *Hodge & Dvorak, 2015*) and *E. grandis* (*Osorio, White & Huber, 2001*), and were slightly lower than those of eucalypts hybrids (*Bouvet, Vigneron & Saya, 2003*), *E. globulus* (*Stackpole et al., 2010*) and *E. saligna* (*Lan et al., 2013*) at the same age. Moreover, the relatively high survivals at SK and TT indicated that the *E. urophylla* clones adapted better to the environments at the two sites. The trends in H growth found at the three sites were similar to those reported for *E. urophylla* × *E. grandis* (*Bouvet, Saya & Vigneron, 2009*) and *E. grandis* reported by (*Osorio, White & Huber, 2001*). H growth increased rapidly till about 3.5 years old before slowing somewhat. Of course, H had not reached a plateau at 7.9 years old in the sites, indicating that these clones involved still had the potential to grow further. The growth rate of DBH over bark decelerated moderately with age beyond about 2.5 years old. The fast-slow trend was similar to several earlier studies in *E. pellita* and eucalypt hybrids (*Leksono, Kurinobu & Ide, 2006*; *Bouvet, Saya & Vigneron, 2009*). This pattern of DBH development might be due to increasing canopy density with the growth of clones up to about 2.5 years old, after which the competition between trees

**Table 9** Mean genetic gain (%) matrices of the top five clones for growth traits across eight years at the three trial sites.

| Age | H (m) | | | DBH (cm) | | | V(m³) | | |
|---|---|---|---|---|---|---|---|---|---|
| | SK | TT | XNJ | SK | TT | XNJ | SK | TT | XNJ |
| 0.5 | 9.7 | 9.7 | 12.9 | 22.0 | 33.7 | 16.7 | 49.0 | 49.8 | 56.0 |
| | (0.30) | (0.30) | (0.35) | (0.41) | (0.20) | (0.23) | (0.42) | (0.31) | (0.36) |
| 1.5 | 16.5 | 16.6 | 9.0 | 5.8 | 12.5 | 9.9 | 28.3 | 42.1 | 22.9 |
| | (0.17) | (0.08) | (0.22) | (0.31) | (0.25) | (0.12) | (0.22) | (0.21) | (0.09) |
| 2.5 | 15.0 | 18.9 | 8.9 | 8.9 | 16.6 | 7.0 | 32.1 | 52.7 | 17.3 |
| | (0.20) | (0.18) | (0.19) | (0.15) | (0.32) | (0.32) | (0.15) | (0.36) | (0.29) |
| 3.5 | 12.5 | 17.8 | 9.2 | 9.2 | 18.2 | 5.2 | 26.9 | 53.2 | 15.5 |
| | (0.08) | (0.23) | (0.17) | (0.07) | (0.31) | (0.22) | (0.08) | (0.35) | (0.18) |
| 4.7 | 8.1 | 16.4 | 3.9 | 7.4 | 19.7 | 5.7 | 20.1 | 53.1 | 6.0 |
| | (0.21) | (0.27) | (0.11) | (0.17) | (0.29) | (0.20) | (0.25) | (0.38) | (0.35) |
| 5.7 | 6.6 | 15.4 | 4.7 | 5.3 | 17.5 | 5.5 | 14.0 | 48.6 | 14.8 |
| | (0.16) | (0.26) | (0.40) | (0.07) | (0.32) | (0.39) | (0.23) | (0.40) | (0.51) |
| 7.9 | 4.9 | 13.3 | 3.9 | 6.4 | 17.5 | 5.3 | 15.4 | 48.7 | 14.0 |
| | (0.08) | (0.21) | (0.25) | (0.20) | (0.31) | (0.24) | (0.20) | (0.37) | (0.31) |

**Notes.**

Note: The select proportion is 25%, and coefficients of variations are listed in brackets. H is the total height, DBH is the diameter at breast height over bark, and V is the individual tree volume over bark.

is significantly increased with competition between tree crowns likely being deleterious to DBH growth of the individual trees (*Neilsen & Gerrand, 1999*), therefore, the individual incremental rate decreased in DBH over bark for *E. urophylla* and its hybrids. Studies on *E. urophylla*, *E. pellita*, *E. nitens,* and *E. camaldulensis* showed that the growth of DBH over bark was likely influenced by the increased spacing (*Medhurst, Beadle & Neilsen, 1999*; *Li et al., 2002*), in which the competition for photo and nutrients between increased spacing trees is decreased. Furthermore, the growth trends for V indicated that the *E. urophylla* clones maintained strong growth potential up to at least 7.9 years old. At the three sites in this current study, V (individual tree volumes) at just 6.0 years old varied from 0.13 to 0.15 m³, which was somewhat higher than those reported in other studies on various eucalypt species at the same age, such as 0.04 to 0.09 m³/tree for *E. urophylla* (*Li et al., 2002*), from 0.09 to 0.11 m³ for *E. tereticornis* (*Varghese et al., 2008*), and were similar to those for *E. camaldulensis* (*Varghese et al., 2008*) and *E. urophylla × grandis* (*Wu et al., 2011*). At 3.5 years old, V varied from 0.05 to 0.06 m³ and this was slightly higher than that reported for *E. saligna* at the same age which ranged from 0.01 to 0.05 m³ (*Lan et al., 2013*).

The values of $H_c^2$ at TT were dramatically higher than those at SK or XNJ, which indicated that the repeatability of *E. urophylla* clones tended to be larger in better environments under strong genetic control. For H, the $H_c^2$ for *E. urophylla* clones varied from 0.940 to 0.973 at TT, which were of similar values to those found in other studies of clones of *E. urophylla × E. grandis* clones (*Retief & Stanger, 2009*; *Li et al., 2014*). For DBH over bark, the $H_c^2$ varied from 0.854 to 0.972 at TT, and for V, which ranged from 0.915 to 0.985 at TT, these results were in agreement with the $H_c^2$ observed in previous studies by *Wu et al. (2015)*, in which $H_c^2$ for V about 0.80 for *E. urophylla*.

It is worth noting that the survivals, H, DBH over bark (after 3.0 years old) and V increments, $H_c^2$ at XNJ were significantly lower than those at SK and TT at the same age, which might be attributed to biotic and abiotic (*e.g.*, soil, climate, and geographical) factors. Firstly, the difference in soil properties may have resulted in the differences. For instance, soil texture, alkaline, clay soil is relatively thin and nutrient-poor at the XNJ site. In contrast, acidic and loamy (or sandy) soils with abundant soil organic matter are relatively thick at SK and TT. Secondly, the difference in previous vegetation may be an abiotic factor. Based on the field investigations, previous vegetation was a plantation of *E. urophylla* and *E. tereticornis* natural hybridization progeny at the SK and TT site, and a plantation of *P. massoniana* at the XNJ site, which may have led to the difference of soil micro-environment. Moreover, though the meteorology conditions at the three trial sites were approximately similar, air humidity and soil humidity at the three sites during the trial may be different due to the actual rainfall and evaporation capacity. Furthermore, differences in actual evaporation capacity due to soil texture, slope direction, and gradient may lead to differences.

Genotype and environmental factors have an important influence on plant growth. Strong genetic correlations for growth traits (H, DBH over bark, and V) between SK and TT indicated that the clones from the two sites were under strong genetic control. However, the weak genetic correlations between SK and XNJ for H, DBH over bark indicated that the G × E interactions were relatively strong (*Berlin, Jansson & Högberg, 2015*), indicating performances of clones relative to each other across these two sites were subject to disturbances from the environment effects (*Kien et al., 2010*). Genetic correlations observed between traits at TT were significantly higher than at the other sites. For V, the genetic correlations for any pair of sites using 1.5 to 3.5-year-old trees, were higher than at 0.5 years old. Therefore, V from 1.5- to 3.5-year-olds could be a reliable selection tool in the early stage, the H at 2.5-years-old was a relatively reliable selection trait index, and DBH over bark from 1.5- to 2.5-years-old was also relatively reliable.

The genetic gains for growth traits at the same age varied between sites in the following order of TT >SK >XNJ, which indicated that *E. urophylla* clones adapted most readily at TT. Overall, the optimum selection age should be between 1.5 and 3.5 years old based on the trends in genetic gains. Some genetic gains at 0.5 years old were the highest among the seven ages, for instance, H at XNJ, DBH over bark at the three sites, and V at SK and XNJ. The reasons for this finding might be explained by: (1) fertilizer had only been applied once at the time of trees being planted; (2) the competition for light, water, fertilizer, and space between trees, or between trees and weeds, was the lowest at the beginning due to the intensive initial site preparation which meant trees were planted into weed-free environments.

The selection in tree age (between 1.5 and 3.5 years old) was similar to the previous studies, which indicated that 2 or 3 years was best in optimum selection efficiency for growth traits of *E. urophylla* (*Kien et al., 2009*; *Pinto et al., 2014*). Based on the optimum age, V is the optimum index of *E. urophylla* selection. Our results were inconsistent with the study of *Berg et al. (2016)*, who indicated that DBH is a sufficient growth measure to use in *E. urophylla* breeding programs. The distinction of *E. urophylla* materials, G × E

interactions, and tree ages tested may contribute to the difference in the index selection of growth traits.

## CONCLUSIONS

*E. urophylla* clones adapt best to the TT site by acidic and loamy soil with a clay content of about 456 g/kg, and the soil depth from the surface to the parent material is about 1.5 m. Moreover, under the specific environment, the growth traits of clones were mainly affected by genetic factors. The optimum selection age for *E. urophylla* clones ranged from 1.5 to 3.5 years old, while V is the optimum selection in terms of growth traits.

## ACKNOWLEDGEMENTS

We are grateful to Yingan Zhu and Wei Wang from the Research Institute of Tropical Forestry, Chinese Academy of Forestry for field and laboratory assistance.

### Funding

This research was supported by the National Key R&D Program of China during the 14th Five-year Plan Period (2022YFD2200203) and the 14th Five-year Plan Period (2023YFD2201003), and the Fundamental Research Funds for the Central Non-profit Research Institution of CAF (No. CAFYBB2022SY017 and CAFYBB2021SY001). The funders had no role in study design, data collection and analysis, decision to publish, or preparation of the manuscript.

### Grant Disclosures

The following grant information was disclosed by the authors:
The National Key R&D Program of China during the 14th Five-year Plan Period: 2022YFD2200203.
The 14th Five-year Plan Period:  2023YFD2201003.
The Fundamental Research Funds for the Central Non-profit Research Institution of CAF: CAFYBB2022SY017, CAFYBB2021SY001.

### Competing Interests

The authors declare there are no competing interests.

### Author Contributions

- Guangyou Li conceived and designed the experiments, performed the experiments, analyzed the data, prepared figures and/or tables, authored or reviewed drafts of the article, and approved the final draft.
- Zhaohua Lu conceived and designed the experiments, analyzed the data, prepared figures and/or tables, authored or reviewed drafts of the article, and approved the final draft.
- Deming Yang performed the experiments, authored or reviewed drafts of the article, and approved the final draft.

- Yang Hu performed the experiments, analyzed the data, authored or reviewed drafts of the article, and approved the final draft.
- Jianmin Xu conceived and designed the experiments, analyzed the data, authored or reviewed drafts of the article, and approved the final draft.

### Data Availability

The raw measurements are available in the Supplementary Files.

### Supplemental Information

Supplemental information for this article can be found online at http://dx.doi.org/10.7717/peerj.18218#supplemental-information.

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
