# Peer review of "Age trends of genetic parameters and genotype-by-environment interactions for growth traits of Eucalyptus urophylla clones in South-China"

_PeerJ, doi:10.7717/peerj.18218_

## Round 0.1 · original submission · Minor Revisions

Dear Authors,

Thank you for your recent revisions. However, both minor corrections are still required before your manuscript can be accepted for publication in PeerJ. We encourage you to address these points, and we look forward to receiving your updated manuscript soon.

Best regards,
Armando Sunny

Reviewer 1 ·

Basic reporting

Eucalyptus is widely planted in southern China. Previous studies have focused on the growth comparison of different clones, and few studies have been seen on the interaction between different hybrids or clones and the environment.
The research paper submitted by Li et al. solves this important scientific problem and provides important reference for future eucalyptus planting strategies.
This research paper has a clear topic, proper references, proper data processing, and clear conclusions.

Experimental design

This study covered 20 clones in three test locations and recorded and analyzed growth data from multiple years. The experimental design was reasonable and the data analysis method was effective, which enabled important conclusions with guiding significance to be obtained.

Validity of the findings

The study reached a brief but very important conclusion, clarifying the suitable cultivation site conditions for Eucalyptus, such as acidic and loamy soil with a clay content of about 456g/kg, and pointed out the optimal selection age. These conclusions have important guiding significance for Eucalyptus cultivation methods.

Additional comments

Table 3 shows the information of the 20 clones tested, including 11 interspecific hybrids. Did this study in this paper only use Eucalyptus urophylla as the research material? Please clarify in the Materials and Methods section.

Reviewer 2 ·

Basic reporting

Your introduction needs more detail. I consider that the articles by Kien et al., 2009; Pinto et al., 2014; van den Berg et al., 2016; and Silva et al., 2018 should be considered for the introduction at lines 74-87 as part of the background in the species.
- Kien, N. D., Jansson, G., Harwood, C., & Thinh, H. H. (2009). Genetic control of growth and form in Eucalyptus urophylla in Northern Vietnam. Journal of Tropical Forest Science, 50-65.
- Pinto, D. S., Resende, R. T., Mesquita, A. G. G., Rosado, A. M., & Cruz, C. D. (2014). Early selection in tests for growth traits of Eucalyptus urophylla clones test.
- Van den Berg, G. J., Verryn, S. D., Chirwa, P. W., & van Deventer, F. (2016). Estimates of genetic parameters and genetic gains for growth traits of two Eucalyptus urophylla populations in Zululand, South Africa. Southern Forests: a Journal of Forest Science, 78(3), 209-216.
- Silva, P. H. M. D., Brune, A., Pupin, S., Moraes, M. L. T., Sebbenn, A. M., & de Paula, R. C. (2018). Maintenance of genetic diversity in Eucalyptus urophylla ST Blake populations with restriction of the number of trees per family. Silvae Genetica, 67(1), 34-40.
On line 121 typo “obviously”.
Text on line 156 is unclear.
On lines 414-416 duplicates citation to Lan et al., 2013.
On lines 492-494 duplicate cites Wu et al., 2011.
For equation (2) the explanation of the term BFjk was omitted.
I consider that there are issues that were not developed in the discussion, such as contrasting the results with other similar studies in the species. For example, van den Berg et al., 2016 indicate DBH as the best measure to use in breeding programs of E. urophylla (diameter is easier and quicker to measure than heights), and in this paper you conclude that it is volume (V).
I thank you for providing the raw data, however your supplementary table Growth_traits.xlsx I consider necessary to add a column with the identification of the clones (clone ID), this allows to estimate the age-age correlations of the measured phenotypes.

Experimental design

No comment.

Validity of the findings

No comment.

·

Basic reporting

This study by Li et al. carried out age trends of genetic parameters and genotype by environment interactions for growth traits of Eucalyptus urophylla Clones in South-China. By comparative analyses among three sites, average phenotypic trends, broad sense repeatability and G×E interactions were estimated. The manuscript submitted has used some new methodologies in modeling genetic parameters of growth traits in Eucalyptus. However a few errors were noticed in the manuscript, and details are given below.

Experimental design

There are issues with the statistical model. The all terms in this combined model do not be tested for significance using a likelihood ratio test, which should be removed from the model if the term does not reach significance. Ultimately, The optimal linear model is used for subsequent analysis. In this paper, it appears that the model may not be correctly applied to the analysis.

Is there a common error variance across all sites? or was a heterogenous R structure used in
the model? I suggest that the block, site variances, clone and site×clone variance components should be listed from different ages?

Validity of the findings

The survivals of E. urophylla clones across eight years at the three trial sites was very low, especially the XNJ trial site, which leads to significant fluctuations in the repeatability. Have you optimized the data in your analysis before conducting the analysis?

Please add the unit for different growth traits in Table 5-7.

Line 249-256 Genetic correlations between growth traits and Type B genetic correlations between the sites should be explained separately based on the table 8.

---

## Round 0.2 · accepted · Accept

Dear Authors,

I am pleased to inform you that all three reviewers agree that the revisions made to your manuscript have been excellent, and it is now ready for publication.

Thank you for choosing PeerJ to share such a compelling and insightful work.

Best regards,
Armando Sunny

Reviewer 1 ·

Basic reporting

I think the revision of the article are appropriate and acceptable for publication.

Experimental design

I think the revision of the article are appropriate and acceptable for publication.

Validity of the findings

I think the revision of the article are appropriate and acceptable for publication.

Reviewer 2 ·

Basic reporting

No comment.

Experimental design

No comment.

Validity of the findings

No comment.

Additional comments

The authors have implemented the requisite corrections as per the recommendations set forth in the preceding review. In my estimation, the work is now suitable for publication.

·

Basic reporting

The paper assess genotype by environment (G×E) interactions related to growth traits and soil factors and identify the suitable genotype for a specific environment. now the paer has more clear data processing and conclusion.

Experimental design

The statistical model has been revised accroding to previous papers and the review. Therefore, it appears that the model is correctly applied to the analysis.

Validity of the findings

No comment.

Additional comments

The previous review and the response of the authors to the queries were found satisfactory.